# Mass Transfer Characteristics of Haemofiltration Modules—Experiments and Modeling

**DOI:** 10.3390/membranes12010062

**Published:** 2022-01-01

**Authors:** Alexandra Moschona, Margaritis Kostoglou, Anastasios J. Karabelas

**Affiliations:** 1Chemical Process and Energy Resources Institute, Centre for Research and Technology-Hellas, 57001 Thessaloniki, Greece; alexmoschona@certh.gr (A.M.); kostoglu@chem.auth.gr (M.K.); 2Department of Chemistry, Aristotle University of Thessaloniki, 54124 Thessaloniki, Greece

**Keywords:** haemofiltration membrane-module, mass transfer, urea convective, diffusive transfer, shell-side local urea-concentration, effective diffusion coefficient in membrane

## Abstract

Reliable mathematical models are important tools for design/optimization of haemo-filtration modules. For a specific module, such a model requires knowledge of fluid- mechanical and mass transfer parameters, which have to be determined through experimental data representative of the usual countercurrent operation. Attempting to determine all these parameters, through measured/external flow-rates and pressures, combined with the inherent inaccuracies of pressure measurements, creates an ill-posed problem (as recently shown). The novel systematic methodology followed herein, demonstrated for Newtonian fluids, involves specially designed experiments, allowing first the independent reliable determination of fluid-mechanical parameters. In this paper, the method is further developed, to determine the complete mass transfer module-characteristics; i.e., the mass transfer problem is modelled/solved, employing the already fully-described flow field. Furthermore, the model is validated using new/detailed experimental data on concentration profiles of a typical solute (urea) in counter-current flow. A single intrinsic-parameter value (i.e., the unknown effective solute-diffusivity in the membrane) satisfactorily fits all data. Significant insights are also obtained regarding the relative contributions of convective and diffusive mass-transfer. This study completes the method for reliable module simulation in Newtonian-liquid flow and provides the basis for extension to plasma/blood haemofiltration, where account should be also taken of oncotic-pressure and membrane-fouling effects.

## 1. Introduction

Haemofiltration in its various modes (i.e., haemodialysis, haemodiafiltration, expanded haemodialysis, etc.), employing hollow fiber (HF) ultrafiltration membrane modules, is a complicated process, which involves mass transfer of relatively small toxic molecules through the membranes, from a non-Newtonian fluid of significant oncotic pressure (i.e., blood) to a counter-currently flowing Newtonian liquid (dialysate). This process is characterized by significant spatial variability of all process parameters across the module, since the composition of both fluids tends to vary because of (a) liquid trans-membrane flow and (b) diffusive and convective type of species transfer through the HF membranes. Indeed, in the currently favored “high flux” HF membranes, there is trans-membrane flow from lumen/blood- to shell-side (“internal filtration”) in the proximal part, and the reverse (“back-filtration”) in the distal part of module [1]. Additionally, there is temporal variability in this process, under the imposed feed-flow rates of the two fluids, which is mainly caused by the tendency of organic macromolecules (notably proteins) to adhere/deposit on the membranes, thus reducing their effective permeability (e.g., [2]), with obvious direct impact on species mass transfer. Significantly, the key haemofiltration performance-parameters, including the sieving coefficient and clearance of the targeted toxic species, are directly affected by the aforementioned complicated spatially/temporally varying phenomena.

Despite the very significant progress made in the field of haemofiltration, particularly during the past two decades, on membrane material properties [3] and introduction of novel (more effective) operating protocols (e.g., [4,5]), serious gaps exist in our knowledge of the HF-module performance as a function of the imposed flow rates at blood- and dialysate-side [6]. The implications of these gaps are clearly reflected, particularly in the inadequate simulation tools available for predicting/determining the performance of various types of modules as well as in the questionable/deficient standards, needed for obtaining representative module specifications and for comparison/selection of modules. For instance, in recent publications, such weaknesses are discussed about the current standards for determination of key module-performance parameters; i.e., the ultrafiltration coefficient KUF [7] and the sieving coefficient [8]. An in-depth study clarifying the serious deficiencies of current standards and practices was published very recently [9]. In parallel, efforts to model and simulate the HF-module performance, through empirical and theoretical models (e.g., [1,10]), have met with modest success (at best), as they have to cope with the aforementioned spatial-temporal process variability and the complicated flow field. The latter is characterized/shaped by the stochastically/irregularly arranged thousands of fibers in the main module-section and by the headers at the two ends of the module [11], where incoming-outgoing fluids are engaged/disengaged. Comprehensive and reliable fluid-mechanical modeling/simulation of this flow-field, based on first principles, is almost impossible at present.

In the present authors’ view, the main issue that has led to this unsatisfactory situation, is related to weaknesses of the methodology employed to characterize the performance of modules. Particularly at the experimental level, the currently employed methods entail some relatively simple tests/protocols that cannot account for all the aforementioned mechanisms and complicated interactions, which lead to the spatial-temporal variability of fluids in a haemofilter/haemodialyzer. To address this issue, a novel systematic methodology is advocated and implemented [12,13], whereby, first, a complete and reliable fluid mechanical characterization of HF modules is performed for (the simpler) Newtonian fluids; this method, combining mechanistic modeling and specific tests [12], has been recently validated [13]. Next, the method is extended to predict/simulate the mass transfer characteristics for Newtonian liquids; this is the objective of the present publication, which involves model extension and experimental validation. In future studies (i.e., next stage of work, to be pursued by the authors), the above two steps will be implemented, employing (the Newtonian) human plasma, thereby introducing the effects due primarily to membrane fouling and oncotic pressure. In the final stage, the method will be extended/adapted to haemofiltation of blood.

In this publication, a summary of the haemofilter/haemodialyzer fluid-mechanical model, and its extension to mass transfer, is presented first. Next, the experimental work is described to study mass transfer of urea, in the typical counter-current flow mode of HF; for this purpose, a specially instrumented test-section is employed, allowing fairly accurate measurements of the local variation of urea concentration on the shell side, under externally imposed constant feed-flow rates. Such data are presented for the first time, to the authors’ best knowledge. The ensuing comparison of the experimental data with model predictions (in addition to model validation) provides valuable insights into the haemofiltration process through a sensitivity analysis involving key process parameters.

## 2. Theoretical Part—Modeling of Transport Phenomena in Haemofilters

### 2.1. Complete Modeling/Characterization of the Flow Field

A model of the experimentally examined process will be outlined here as a tool for data analysis, toward the development of a reliable haemofiltration-process simulator. The flow and concentration fields in the active cylindrical-section of haemofilters are complicated mainly due to geometric complexities of the shell-side geometry, arising from the random arrangement of the numerous fibers [14]. This problem is typically overcome by considering an “average” unit cell, consisting of a single fiber and an annular Happel-type cell around this fiber to represent the shell side [15]. There are several models in the literature for the simulation of the flow field either of two-dimensional type (where Computational Fluid Dynamics is employed) [10] or of one-dimensional nature (one-dimensional mass and momentum balances employed) [16]. The latter type of models relies on theoretical values of friction factors using expressions from the literature and nominal geometrical values for the device. However, it was recently shown [12] that the existing models are clearly deficient because they ignore the inertia pressure losses occurring in the headers, at the inlet and outlet of the active section of the module. A procedure has been developed to obtain directly from experimental data (of pressure drop versus flow rate, in specific operating modes) the friction coefficients, thus alleviating the need to invoke theoretical relations. Having estimated the friction factors and the membrane permeation coefficient, the complete one-dimensional profile of flow rates along the unit cell, during counter-current operation of the device, can be obtained as follows:(1)$$Q_{f}=-\frac{Q_{fo}}{A_{f}}[\frac{A_{f}}{A}(-c_{1}e^{-Az/L}+c_{2}e^{Az/L})+c_{3}]$$
(2)$$Q_{s}=\frac{Q_{so}}{A_{s}}[-\frac{A_{s}}{A}(-c_{1}e^{-Az/L}+c_{2}e^{Az/L})+c_{3}]$$
(3)$$Q=\frac{Q_{fo}}{L}(c_{1}e^{-Az/L}+c_{2}e^{Az/L})$$
(4)$$c_{3}=(\beta +\gamma -1){\left(\frac{1}{A_{f}}+\frac{1}{A_{s}}\right)}^{-1}$$
(5)$$c_{2}=[A\gamma +(1+\frac{c_{3}}{A_{f}})A(e^{-A}-1)]{\left(e^{A}-e^{-A}\right)}^{-1}$$
(6)$$c_{1}=c_{2}+(1+\frac{c_{3}}{A_{f}})A$$

β = Q_so_/Q_fo_, γ = Q_UF_/Q_fo_, A_f_ = f_f_KL^2^, A_s_ = f_s_KL^2^, A = (A_f_ + A_s_)^0.5^

Here, Q_s_ and Q_f_ are the dialysate- and blood-side flow rate, respectively, Q is the local transmembrane flow rate per unit length, Q_UF_ is the net ultrafiltration rate, f_f_ the lumen friction factor, f_s_ the shell friction factor, K the membrane permeance, L the length of HF active-section and z the distance along the active-section. The subscript “o” designates inlet conditions.

The wall Reynolds number Re_w_ is computed as v_w_R/ν (R is either the inner fiber radius for lumen-side or the outer fiber radius for shell-side), v_w_ is the corresponding normal to wall velocity and ν the liquid kinematic viscosity. The values of Re_w_ for the present experiments (and for haemofiltration in general) are of order 10^−3^. Under these conditions, the flow field in the unit cell can be decomposed into two flow fields, an axial and a radial one, where a one-sided coupling is considered [17]. Indeed, the axial problem, with solutions represented by Equations (1)–(6), is independent from the radial one. However, the latter depends on the axial problem and can be described by the following expressions, as shown elsewhere [18]:

For 0 ≤ r ≤ R_o_,
(7)$$u(z,r)=2U(z)(1-{\left(\frac{r}{R_{o}}\right)}^{2})$$
(8)$$v(z,r)=v_{w}(z)(\frac{2r}{R_{o}}-{\left(\frac{r}{R_{o}}\right)}^{3})$$

For R_1_ ≤ r ≤ R_c_,
(9)$$A_{c}=\frac{1}{1-4T^{2}+3T^{4}-T^{4}\ln(T^{4})}$$
(10)$$u(z,r)=U_{c}(z)[1-{\left(\frac{r}{R_{1}}\right)}^{2}+T^{2}(\ln\left(\frac{r^{2}}{R_{1}^{2}}\right)-1)]{\left[\frac{1-T^{2}}{2}+\frac{T^{4}\ln(T^{2})}{T^{2}-1}\right]}^{-1}$$
(11)$$\begin{array}{l} v(z,r)=v_{wc}(z)\frac{-2A_{c}R_{1}}{r}[-\frac{1}{2}{\left(\frac{r}{R_{1}}\right)}^{4}+{\left(\frac{r}{R_{1}}\right)}^{2}+T^{2}{\left(\frac{r}{R_{1}}\right)}^{2}(\ln{\left(\frac{r}{R_{1}}\right)}^{2}-1)+\\ +\frac{T^{4}}{2}-T^{2}-T^{4}(\ln(T^{2})-1)] \end{array}$$

Here R_o_ and R_1_ are the inner and outer fiber radius, respectively. The velocities U, U_c_ designate the mean axial velocities in lumen and shell-side, respectively, which can be determined by dividing the flow rates Q_f_, Q_s_ by the corresponding cross-sectional areas. The wall flux v_w_(z) is computed as Q/(2πR_o_), whereas the wall flux v_w_c(z) is given as Q/(2πR_1_). The variable T is simply the ratio R_c_/R_1_, where R_c_ is the outer radius of the unit cell on the dialysate/shell side. The value of R_c_ can be computed either by using the nominal volume fraction of dialysate side ε and the relation ε = (R_1_/R_c_)^2^ or (even better) by the theoretical relation of the experimentally found fs based on flow in the Happel unit cell [12]. The latter approach is followed here.

### 2.2. Modeling Mass Transfer in Haemofilters

The above analysis is important as it allows to reconstruct in closed-form the complete two-dimensional flow field directly from experimental data, with no need to take a Computational Fluid Dynamics approach. However, still a convection-diffusion mass balance of the solute in 2-dimensions and for three domains (i.e., two channels and interior of membrane) must be solved. There is a particular limit (i.e., *no diffusion of the solute*) for which the above task can be avoided and a very simple solution for the solute fraction leaving the blood side can be derived. In this case, the solute follows the liquid in the blood entrance region, but no solute transfer takes place in the opposite direction, since the dialysate is free of solute in its entrance region. Thus, solute is transferred from the blood stream only in the proximal region where liquid trans-membrane flux occurs from lumen to dialysate. The point z′ which indicates the position of trans-membrane flow reversal can be determined by the relation Q(z′) = 0. The substitution in Equation (3) leads to z′ = L(−c_1_/c_2_)^0.5^. Furthermore, substitution of this expression for z′ in Equation (1) allows determination of blood-side liquid-fraction transferred to the dialysate-side in the proximal region as follows: [1 − Q_f_(z′)/Q_fo_]. This expression also corresponds to the *minimum* possible fraction of solute removed (only by convective transfer) for a specific set of flow conditions.

The development is pursued next for the case of the *non-zero diffusion coefficient of the solute.* Having determined the flow field, the concentration field of the solute is of interest. The need for cumbersome detailed numerical solution of the two-dimensional problem is questionable, since it is based on the major approximation of an annular dialysate unit cell. Other issues, as the axial convection and diffusion (e.g., [19]) in the membrane, can be neglected due to high aspect ratio of the membrane. To overcome the required (and rather unnecessary) computational effort, several approximate one-dimensional models have been developed in the literature, of possible usefulness to this study; therefore, this direction will be followed here.

The solute is transferred by diffusion and convection at both sides (lumen and shell) of the membrane and through the membrane. The solute balances in the two sides are handled through the use of the mass transfer coefficients. Here, G denotes the solute mass transferred from one side to the other per second and per meter of fiber length (units: kg/m/s). The solute concentrations on the two sides are denoted as C_B_ and C_D_, respectively (“B” for blood and “D” for dialysate side, respectively). The mass balances are:(12)$$\frac{dQ_{f}C_{B}}{dz}=-G$$
(13)$$\frac{dQ_{s}C_{D}}{dz}=G$$

To proceed, furthermore, an expression for the quantity G is needed. The general form of this expression, used in the literature, includes contributions from both convection and diffusion [14] as follows:(14)$$G=QC_{a}+K_{D}(C_{B}-C_{D})$$
where C_a_ is a characteristic concentration (between C_B_ and C_D_) which is considered representative of the convection contribution to transmembrane mass transfer. The effective mass transfer coefficient K_D_ (units: m^2^/s) accounts for mass transfer in channels and for diffusion through the membrane. Many studies modeling the haemodialysis process use the so-called Kedem–Katchalsky (KK) equation for transport of non-electrolyte solutions through the membrane [20]. This equation (of thermodynamic origin) has found extensive application for many types of membrane processes. According to this approach, the characteristic value C_a_ is simply the mean value of C_B_ and C_D_, i.e., C_a_ = (C_B_ + C_D_)/2 (KK1 version). Another version of the KK equation considers C_a_ as the logarithmic mean of C_B_ and C_D_ (KK2 version) [21].

A different approach has been followed by Zydney and co-workers [22,23], who solved analytically the diffusion-convection equation in the porous membrane. This model also takes into account the sieving coefficient which can be different from 1 for solutes of large molecular size. The case of a sieving coefficient different than 1 will not be considered in the present work to retain compatibility between the relations for C_a_ and because this is the actual value for the toxic substance considered here (urea). The expression of the Zydney group, with sieving coefficient equal to 1, after algebraic manipulations, can be transformed to the following expression (Z model):(15)$$C_{a}=\frac{e^{\gamma }}{e^{\gamma }-1}C_{B}-\frac{1}{e^{\gamma }-1}C_{D}$$
where the parameter γ = Q/K_D_ denotes the relative strength of convection to diffusion. In order to demonstrate the difference of the three approaches (KK1, KK2 and Z), the ratio C_a_/C_B_ as a function of the ratio C_D_/C_B_ is presented in Figure 1.

It is noted that the lower bound to the characteristic concentration C_a_ is provided by the KK1 model. This is equivalent to the Z model in the limit γ << 1 (i.e., diffusion dominated process). According to the Z model, as the influence of convection increases (γ increases), the characteristic concentration moves from the average to C_B_. In particular, for γ = 3, the Z model gives results very similar to KK2. As γ further increases, C_a_ continues to increase and, for γ >> 1 (convection dominated process), C_a_ tends to C_B_. The above analysis makes it clear that KK1 and KK2 correspond to specific cases of the Z model, which is much more general; therefore, the latter will be employed in the present haemofiltration studies. It is stressed that, at least for solutes of relatively small molecular weight (as urea), the mass transfer model of the haemofiltration is similar to that of an ultrafiltration process (where species convection and diffusion occur), but different from the models for forward [24] and reverse [25] osmosis where solute rejection dominates due to tighter membranes.

In order to complete the analysis, an expression is needed for the overall mass transfer coefficient K_D_. The original relations for G, described above, assumed diffusional mass transfer to occur only through the membrane, and they employ the diffusional permeability concept. Accounting for the convective diffusion mass transfer at both membrane-side channels leads to the addition of the term “modified” to the corresponding equation. Here, instead of using the concept of diffusional permeability, the development will be made in terms of bulk solute diffusion coefficient and membrane structure. The effective diffusion coefficient of the solute in the membrane, denoted as D_e_ (units: m^2^/s), is related to the bulk diffusion coefficient D of the solute in the liquid through D_e_ = λD, where λ is a dimensionless *diffusion hindering factor* which depends only on the membrane structure. For the latter, a usual relation is λ = φ/τ, where φ is the membrane porosity and τ the membrane tortuosity (usually greater than 1 for polymeric membranes). The solution of the one-dimensional diffusion problem in the membrane leads to the following relation:(16)$$K_{D}=2\pi {\left(\frac{1}{R_{0}h_{in}}+\frac{1}{R_{1}h_{out}}+\frac{1}{\lambda D}\ln(\frac{R_{1}}{R_{0}})\right)}^{-1}$$
where h_in_, h_out_ are the inner (cylindrical) and outer (annular) channel mass transfer coefficients. It is noted that the usual approach in the literature is to neglect the influence of the transverse (due to porous wall) flow field on the mass transfer coefficients. Such an approach is validated for the pressure drop relation, but it is questionable for the mass transfer coefficient [26]. Nevertheless, the usual approach is followed here; i.e., the mass transfer coefficient on the two sides are considered functions of the coordinate z along the flow. There are several relations for the cylindrical channel in the literature, but most of them apply to average coefficients along the flow and not to local ones. To overcome this problem (also considering that the contribution of the channel mass transfer coefficients is not the dominant one), an approximating technique is employed; i.e., the mass transfer coefficient is estimated as follows, using the generalized interpolation technique of Churchill [27] that involves the asymptotic value and the Leveque solution:h = [3.66^5^ + (CN_Gr_^1/3^)^5^]^1/5^(17)

Here, N_Gr_ is the local Graetz number, defined as 4UR_o_^2^/(Dz), and U the local cross sectional average velocity in the lumen; the parameter C has the value 1.07 for the local mass transfer coefficient [28]. The same expression can be used for the dialysate channel coefficient h_out_ substituting the appropriate hydraulic radius in place of R_o_, whereas U, z refer to the outer channel cross sectional average velocity and to the distance from the flow-entry location, respectively. The countercurrent problem is a boundary value one, so a concentration value for dialysate at z = 0 is assumed and an iterative procedure is followed until convergence to the specified value of dialysate inlet concentration at z = L (C_Do_).

The efficiency of the device can be expressed in terms of *fractional clearance* defined as:C_L_ = [Q_Bo_C_Bo_ − Q_B_(L)C_B_(L)]/(C_Bo_Q_Bo_)(18)

This efficiency index, for specific conditions on the blood- and dialysate-side, is simply the mass fraction of solute removed from the blood-side stream, under steady conditions. For comparison with model predictions, data of the corresponding *percent clearance* are reported in this study. It should be also pointed out that, in haemofiltration literature, including the relevant ISO standard [29], *clearance* is alternatively defined as
(19)$$\;\;\;K_{{\mathrm{CL}}^{\;}}=\left(\frac{{\;C}_{\mathrm{Bo}}-{\;C}_{B\left(L\right)}}{C_{\mathrm{Bo}}}\right){*Q}_{\mathrm{Bo}}+\frac{C_{B\left(L\right)}}{C_{\mathrm{Bo}}}{*Q}_{\mathrm{UF}}$$

The above expressions of solute clearance are simply related as follows:K_CL_ = C_L•_Q_Bo_(20)

As noted in Section 2.2, in case of zero solute diffusivity, the fractional solute clearance is given as C_L_ = 1 − Q_f_(z′)/Q_fo_, where the point of zero local transmembrane flux z’ is obtained by solving the transcendental equation Q(z’) = 0. This clearance estimate corresponds to the *minimum value of C_L_*, which is a monotonically increasing function of solute diffusivity.

## 3. Experimental Part

### 3.1. Materials Used—Instrumented Module/Haemofilter

Urea (p.a.), purchased from Penta Chemicals (Czech Republic), was used to prepare a stock solution (5 g/L in deionized water), to be used as feed solution on the blood side. All experiments were conducted using deionized water on the dialysate side. Hemodialysis simulation tests were performed at 25 °C, while the blood-side feed solution was stirred, to eliminate concentration gradients during the experiment.

All experiments were conducted by employing a commercially available *high flux hemodialyzer* of effective membrane surface area 1.9 m^2^ (Elisio 19H, Nipro Medical Corporation [30]), comprising Polynephron™ (polyethersulfone) hollow fibers, with special embedded sampling ports on the shell side, described as follows. Eight small holes were drilled in the shell carefully, to avoid “injuring” the hollow fibers. As shown in Figure 2**,** these holes were arranged in a zig-zag manner to minimize probe interference; the distance between the cross-sectional planes/locations of neighboring holes was 3 cm. In each hole, a short and thin hypodermic needle (size 23G × 1″) was glued, entering inside the fiber bundle by ~1 cm. Parts of disposable plastic syringes were used as plugs for sealing the needles, when not in use for fluid sampling. The process of sampling involved unplugging a port, fixing a syringe and withdrawal of liquid, at a very small rate to minimize disturbance of the shell-side flow field.

### 3.2. Experimental Set up—Operating Modes

The experimental set up for in vitro haemo-catharsis (HC) studies was equipped with two feed vessels, at the inlets of blood- (capacity 2500 mL) and dialysate-side (5500 mL). Two magnetic drive gear pumps (flow rate range 0–1000 mL/min, type MS204, Fluid-O-Tech) were used to feed blood- and dialysate- side solutions, at module top and bottom, respectively. The experimental set up was equipped with four precision pressure transducers (range 0–15psi, type A-10, Wika) installed at the inlet and outlet, at each side, of the module. Blood inlet and dialysate inlet and outlet flow were monitored using three flowmeters (101-Flo-Sequate data, McMillan Co., Cesar Chavez, San Francisco, CA, USA). Additionally, four (4) needle valves were installed at the inlet and outlet of both loops to adjust the flowrates. Moreover, the experimental unit was equipped with Programmable Learning Controller, PLC (CMT Series, Weintek) enabling continuous adjustment, monitoring and recording of all operating parameters (pressures, flow-rates). Data were continuously monitored and recorded every 30 s.

The experimental study is divided into two parts, dealing sequentially with: (1) the determination of *fluid mechanical parameters* of the Elisio 19H dialyzer [12,13], and (2) the collection of adequate data, enabling the study (and validation of model) of the *mass transfer performance* of this dialyzer. Deionized water on the dialysate side and urea solution on the blood-side (with concentration 5 g/L) were used as feed solutions for these two types of experiments.

The first experimental part comprises experiments described in detail in previous publications [12,13]. Specifically, special operating Modes #3 and #4, using deionized water, were performed for determination of the module fluid-mechanical parameters, which were subsequently used in the mass-transfer modelling/simulation. The flow rates tested were 200, 250, 300, 350, 400, 450 and 500 mL/min in Mode #3, and 200, 300, 400, 450, 500, 550 and 600 mL/min in Mode #4.

The second experimental part, dealing with the assessment of the mass transfer performance, was carried out in the common counter-current flow mode. Urea solution was pumped into the hollow fibers from the top of the module, while deionized water was fed into the shell side of the module at the bottom. Three sets of flow rates (cases I, II, III) were tested with this operating mode; i.e., 200/300 mL/min, 250/400 mL/min and 300/500 mL/min, for blood/dialysate flow-rates, respectively; the system was operated at a constant temperature of 25 °C, maintained by a thermostatted bath.

In the beginning of each experiment, deionized water was pumped through both lumen and shell side, for sufficient time (approx. 10 min), to flash the dilute sodium meta-bisulfite solution (used to prevent micro-organism/biofouling development) and to remove air, which may be trapped in the system. Adjustment of desirable flow rates followed using the test fluids on each side. Next, samples were collected from the feed solution of blood-side, from both outlet ports of blood and dialysate-side as well as from the eight special dialysate sampling ports, in the shell. Steady conditions prevailed during sampling, as shown in typical recordings of blood- and dialysate-side inlet flow rates (Appendix A). The concentration of urea, contained in these samples, was determined using a Total Nitrogen Analyzer (TOC-LCSN/TNM-L analyzer, Shimadzu Corporation), through a calibration curve of the total nitrogen amount versus urea concentration. Urea clearance was calculated by employing the above-mentioned expressions (Equations (18) and (19)), using (for each set of conditions) the measured inlet and outlet blood- and dialysate-side concentrations and flow rates. These data as well as the local measurements of dialysate urea-concentration along the module were used for assessing the mass transfer performance of module, in comparison with the corresponding values from modelling/simulation. All experiments that lasted at least 15 min (under steady conditions) were performed in triplicate for each set of tested flow rates.

## 4. Experimental Results—Comparison with Model Prediction

### 4.1. Fluid Mechanical Module Characteristics

Figure 3a,b depict the measured characteristic pressure differences, corresponding to the special operating protocols designated as Modes #3 and #4, respectively. The respective experimental data, plotted in these figures, are listed in the Appendix A. The key fluid-mechanical parameters of this particular module are determined by fitting appropriate lines to these data and following the procedures discussed in detail in [12,13]. The nonlinearity in the variation of these pressure differences is indicative of the (inertia dominated) pressure drop in the two headers of the module. The determined parameter values, for the specific instrumented module, are listed in Table 1. It is noted that the system is over-specified. The six values for the slopes at zero flowrate, of the curves representing the pressure data (see Figure 3), are fitted by only three parameters (K, f_f_, f_s_). The basic assumption is that these parameters are uniformly distributed along the device. If this is not the case, the outcome of the fitting denotes the best possible representative value. The deviation between model and experiment in some points is a combination of experimental uncertainty and deviation from uniformity. The coefficient of determination R^2^ of the fitting is in all cases larger than 0.97.

By employing the model expressions summarized in Section 2.1, the flow field of the module, for counter-current flow of Newtonian fluids, can be fully reconstructed. Figure 4a,b depict the axial variation of lumen-side flow rate Q_f_ and shell-side flow rate Q_s_, respectively, for the three cases studied, which involve three combinations of Q_f_ and Q_s_ feed flow rates. The effect of internal filtration and back-filtration on the spatial variation of these quantities is evident. The point z′ discussed before is the location of the combined minimum of Q_f_(z) and Q_s_(z) curves.

The axial variation of local trans-membrane flow rate Q presented in Figure 5, which is directly related to the flow rate profiles of Figure 4, is of significant interest, as it clearly depicts the regions of the active module section corresponding to “internal” filtration (blood- to dialysate-side) and to “back-filtration” in the reverse direction. Indeed, the axial distance z′ ≈ 0.145 m, [i.e., (z/L) ≈ 0.52], where Q = 0, is the demarcation point of these regions, in all three cases. It should also be noted that the Q axial variation is nonlinear (contrary to the simplifying assumption of linearity often made, e.g., [1]) and that there is an inflection point roughly at z′ ≈ 0.145 m. It is also significant to note that the magnitude of the transmembrane flow rate Q in a broad region, around the point z′ ≈ 0.145 m, is quite small. This means that, in this region, diffusion always has an important contribution to solute transfer irrespective of how dominant convection is near the ends of the active module section.

### 4.2. Mass Transfer Module Performance/Characteristics—Method Validation

#### 4.2.1. Experimental Results

Table A1 (Appendix B) includes all measured inlet and outlet stream (blood- and dialysate-side) process parameters as well as the net ultrafiltration rate Q_UF_ and urea clearance, for all three cases (I, II, III) studied. The local urea concentration measurements, made in triplicate (tests a, b, c) on the shell side of the module, using special probes, are presented in Figure 6, Figure 7 and Figure 8, for the three cases (I), (II), (III), respectively. The measured concentration at the point (z/L) = 0 corresponds to the measured exiting dialysate-side stream concentration C_D_. The data are very consistent, with a deviation from the mean within ±0.05 g/L in almost all cases, and they are characterized by a fairly *smooth axial variation* from (z/L) = 0 to 1.0. However, this smooth variability of C_D_ does not reflect the aforementioned significant changes of trans-membrane liquid flow rate Q, occurring around (z/L) ≈ 0.50, as one might have expected. Moreover, these C_D_ profiles cannot be further assessed, even qualitatively, unless a reliable/ comprehensive process model is available, as the one presented and validated here. It should be added that mass-balance closure checks for all runs (based on the data listed in Table A1) are very satisfactory (deviation within 2%), which indicate the accuracy of these data. An example of such a closure check is included in the Appendix A.

#### 4.2.2. Data Interpretation/Assessment—Model Validation

The local dialysate-side concentration data (Figure 6, Figure 7 and Figure 8) can be assessed, through the fitting of model expressions, outlined in Section 2.2, thus obtaining new insights into the mass transfer process. The input parameter values, for the mass transfer simulations, are listed in Table 2. Parameter values obtained in the preceding fluid-mechanical characterization stage (i.e., K, f_f_, f_s_) are an essential input to achieve a realistic mass-transfer simulation. In these calculations, the only (adjustable) parameter, needed to obtain the overall “best-fit” to the experimental data, is the diffusion hindering factor λ, which is related to the effective diffusion coefficient of the solute in the membrane, D_e_ = λD. This factor depends on the (unknown) particular physico-chemical membrane properties; i.e., porosity, tortuosity, solute molecule size to pore diameter ratio, etc. [31]. The rest of the parameter values in Table 2 are the nominal ones reported by the module manufacturer [30], or estimated from direct measurement (i.e., L). It is noted that the value of urea diffusivity in aqueous solutions at 25 °C, listed in Table 2, is within ±3% of the respective values reported in literature [32,33]. The feed flow rates Q_B_, Q_D,_ the net transmembrane flux Q_UF_ and the feed concentrations C_B_, C_D_ are basic process inputs.

Using straightforward trial and error computations, an overall best fitting is obtained of the model to the local concentration profiles at dialysate-side, C_D_ and to inlet-outlet blood-side C_B_ concentrations, for hindering factor value *λ ≈ 0.095 (±0.005).* In Figure 9, Figure 10 and Figure 11, the comparison is presented between experimental data and the theoretical model simulations. In the depicted model predictions, two small values of λ (i.e., λ = 0.090 and 0.100) are employed. The very small difference of these λ values (affecting computations) reveals that the blood-side model predictions are apparently more sensitive to this parameter than the predictions for dialysate-side concentration.

The comparisons in Figure 9, Figure 10 and Figure 11 show a very good agreement overall. In fact, the agreement at the relatively smaller flow rates (Case I, Figure 9) is excellent. In the other two cases II and III (Figure 10 and Figure 11), some deviations of model predictions from data (regarding the C_D_ profile) are observed for z/L smaller than ~0.5, i.e., in the “internal filtration” region of the module. However, it is difficult to attribute this rather small model under-prediction to either experimental error or model weakness, or both. It is reminded that (as in the case of hydrodynamic parameters) the parameter λ may also be non-uniform along the active region, contrary to uniformity considered here. Nevertheless, the most important result of the comparisons in Figure 9, Figure 10 and Figure 11 is that both data and model predictions show a significant urea concentration C_D_ in the distal half of module (i.e., at z/L greater than ~0.5) due to diffusion. In that part of the module, “back-filtration” occurs, which is characterized by significant convective trans-membrane flow from dialysate- to blood-side; i.e., in a direction opposite to that of urea concentration gradient which drives diffusion.

Moreover, the concentration C_D_ tends smoothly to zero with increasing (z/L) toward 1.0, where this convective trans-membrane flow attains its highest values, as shown in Figure 5. It is also noted that, as the feed blood-side concentration C_B_ is a usual input parameter, comparison of data and predictions on the blood-side is only made with the measured exit C_B_ concentration, which is satisfactory.

Comparison of model predictions, with experimentally determined urea clearance for the three studied cases I, II, III, is presented in Table 3**.** The two values of parameter λ (0.09 and 0.10) used in model predictions show some sensitivity. For a fitting value λ = 0.095, there is a small deviation of prediction from measured clearance C_L_, within ±5%, which is in line with the small deviation of the predictions from measured axial profiles, as shown in Figure 9, Figure 10 and Figure 11. The experimentally determined clearance values C_L_ (%) and K_CL_ (in mL/min), together with the other relevant data, for all tests, are listed in the Appendix B (Table A1).

## 5. Discussion

It is evident that there is a very complicated interaction, along the HF module, of the two basic mechanisms (i.e., convective and diffusive solute transfer), which determines the observed net outcome of the mass transfer process. This complicated interaction prevails particularly in the case of high flux HF modules, where rather strong bi-directional (“internal”- and reverse/“back”-filtration) convective trans-membrane liquid flows occur. As recently acknowledged [6], there is poor understanding of these interactions that hinders our ability to develop reliable process simulation tools needed in optimizing the design and operation of the HF modules. The comprehensive methodology, demonstrated herein for Newtonian fluids, is considered a significant step toward improving this unsatisfactory situation.

Regarding methodology, the novelty of the present approach is that it tackles *first* effectively the reliable and complete fluid-mechanical characterization of the HF module [12]. The realistic assumption is made here that the simultaneously occurring (solute) mass transfer process does not affect the flow fields involved. In the *next step*, utilizing the already determined fluid-mechanical parameters, the mass transfer process in the HF module is completely described. This methodology is demonstrated herein for Newtonian fluids on both the blood- and dialysate-side.

Regarding new insights into the relevant mechanisms, this work (combining detailed experimental data and a mechanistic model) allows for directly assessing the contribution of diffusion compared to convective transfer. This is seldom, if ever, done in the related literature so far. As shown in Figure 9, Figure 10 and Figure 11, the contribution of diffusion is quite significant even in the distal part of module, where there is rather strong liquid transmembrane flow (“back”-filtration) in a direction opposite to that of concentration gradient driving the diffusion process.

The membrane-diffusion hindering parameter λ, in connection with the measured axial dialysate-concentration C_D_ profiles, is convenient for quantitatively assessing the aforementioned contribution of diffusion to the overall mass transfer. In Figure 12 and Figure 13, for the blood-side and dialysate-side, respectively, the measured C_D_ profiles for Case I, are contrasted with theoretical profiles representing several levels of contributions of solute diffusion, by varying the parameter λ within three orders of magnitude (i.e., from 10^−2^ to 1). Similar comparisons for cases II, III are provided in the Appendix A. These figures clearly show that the two “limiting” cases of almost negligible diffusive contribution (λ = 0.01) and of dominant contribution (λ = 1), compared to the data, are not representative of the mass transfer process actually occurring in the HF module. In fact, parameter λ values of intermediate magnitude (i.e., λ ≈ 0.1) appear to characterize the diffusive contribution under the conditions studied here. Indeed, as shown in the preceding section, a *value*
*λ ≈ 0.095 provides a near optimal fit* of model to the present data, obtained with a particular type of HF membrane/module [30], in the range of tested flow rates. Figure 12 and Figure 13 show the “leveling” of the concentration profiles (characterized by limited change) for λ values of order unity; this is due to the fact that, as λ increases, the dominant mass-transfer resistances are those in the channels and not in the membrane (see Equation (16)). Therefore, the total mass transfer coefficient K_D_, for values of hindering factor that tend to λ ≈ 1, is independent of λ, thus approaching the K_D_ value determined by channel coefficients.

Figure 14 clearly depicts a similar effect of diffusion hindering parameter λ values on urea clearance, for the tested case I. Indeed, for values that tend to λ ≈ 1, the reduced resistance to mass transfer by the membrane leads to an asymptotic (maximum) clearance value of 100%, for the studied urea species of small molecular weight. It is noted that the model with the particular set of parameters derived here can be directly used for the calculation of clearance of any other non-sieved molecule by simply changing the value of bulk diffusion coefficient D.

In closing the discussion on solute diffusion, it should be noted that the diffusion hindering parameter λ is (by necessity) a “global” type of parameter, representative of the entire membrane fiber-bundle, which is influenced by (and accounts for) the possible local variability of fiber material/structure throughout the entire module. Therefore, λ (and in turn the effective solute-diffusivity in the membrane) should be experimentally determined for each membrane/ fiber type used in haemofilters.

The overarching goal of this research is to develop the methodology presented herein for application to the currently employed hemofiltration protocols in medical practice. As a first step, simple Newtonian liquids and low molecular weight substances are employed in these studies in order to investigate the phenomena involved and to identify the key intrinsic hydrodynamic and mass transfer parameters of the membranes. In the next steps, additional procedures will be investigated/developed to cope with additional effects, such as oncotic pressure, and to study processes such as membrane fouling, by employing the already determined parameters through the suggested here experimental procedures. The authors are currently pursuing these goals by employing human plasma in haemofiltration experiments.

## 6. Conclusions

A study is presented of the mass transfer characteristics in a haemofiltration type of experiment, employing Newtonian liquids, which combines detailed measurements of local concentration of the transferred species (i.e., urea) on the shell-side with appropriate mechanistic modeling. Aqueous urea solutions are fed on the blood-side, using three combinations of blood- and dialysate-side flow rates. The approach taken is novel and comprehensive, in that, *first,* the key fluid-mechanical parameters (of the particular HF membrane module) are determined, by employing a recently developed method. Thus, complete and reliable fluid-mechanical characterization of the HF module is obtained, including the flow rate profiles at lumen- and shell- side as well as the regions of “internal-” and “back”-filtration in the module. Next, using these fluid mechanical parameters (i.e., membrane permeance, friction coefficients at lumen and shell-side), the new detailed experimental data (i.e., concentration profiles) are assessed by a realistic theoretical model. A diffusion hindering factor λ, of solute species through the membrane, is the only adjustable parameter in the data assessment and model validation presented herein. This parameter λ represents an (unknown a priori) intrinsic membrane/solute property that has to be determined experimentally, as demonstrated here.

By fitting the model to the data, a value of λ is obtained that leads to a very satisfactory agreement overall, of predictions with measurements, that include the concentration profiles of urea on the shell side, the exit blood-side concentrations and the urea clearance, under the tested conditions. Significant new insights are also gained by assessing the data, regarding the relative contributions of convective and diffusive components to total mass transfer; i.e., it is evident that diffusive mass transfer (for the small urea molecules tested) is quite important even in the distal part of the module, where there is transmembrane liquid flow in a direction opposite to the concentration gradient that drives diffusion. A parametric study reveals the magnitude of λ values associated with either diffusion- or convection- dominated mass transfer processes.

In the next stage of further work, the approach successfully demonstrated here, for simulating the haemofiltration process for Newtonian fluids, will be further developed to address the complications introduced by oncotic pressure and fouling phenomena associated with the use of human plasma, and later with blood.

## Figures and Tables

**Figure 1 membranes-12-00062-f001:**
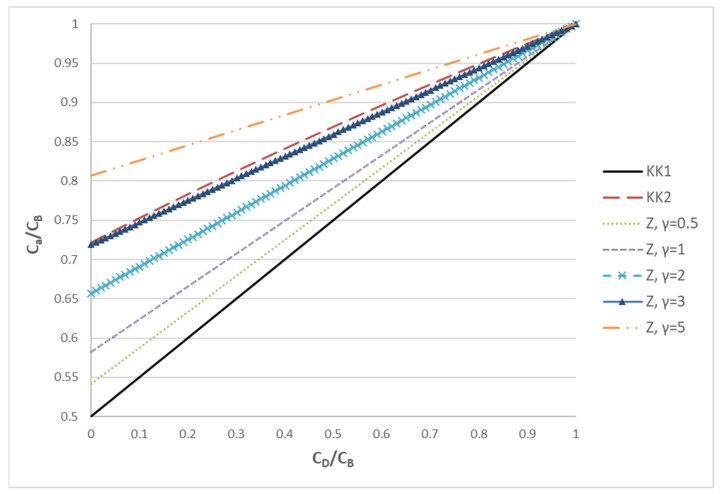
The ratio of C_a_/C_B_ plotted versus the ratio C_D_/C_B_ in order to assess the trends of three different approaches (KK1, KK2, Z) for estimating the characteristic concentration C_a_.

**Figure 2 membranes-12-00062-f002:**
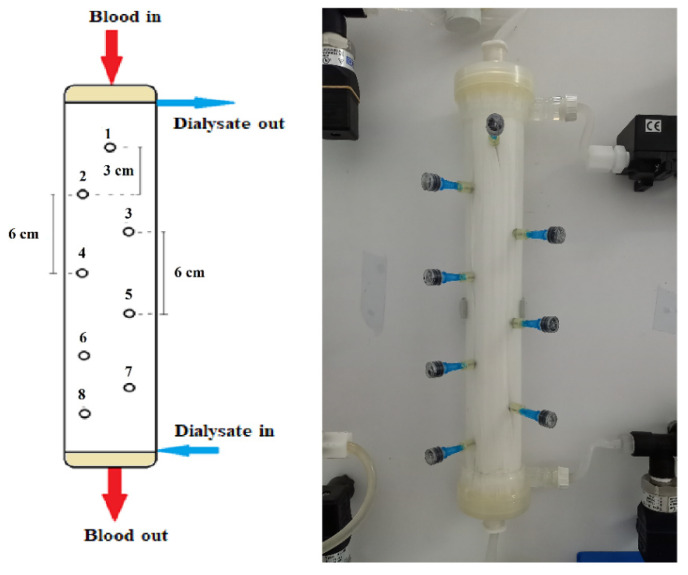
Schematic and image of the instrumented haemodialyzer, showing geometrical details of the eight (#1 to #8) sampling ports/locations. Blood- and dialysate-side feed-fluids pumped at the top and bottom, respectively.

**Figure 3 membranes-12-00062-f003:**
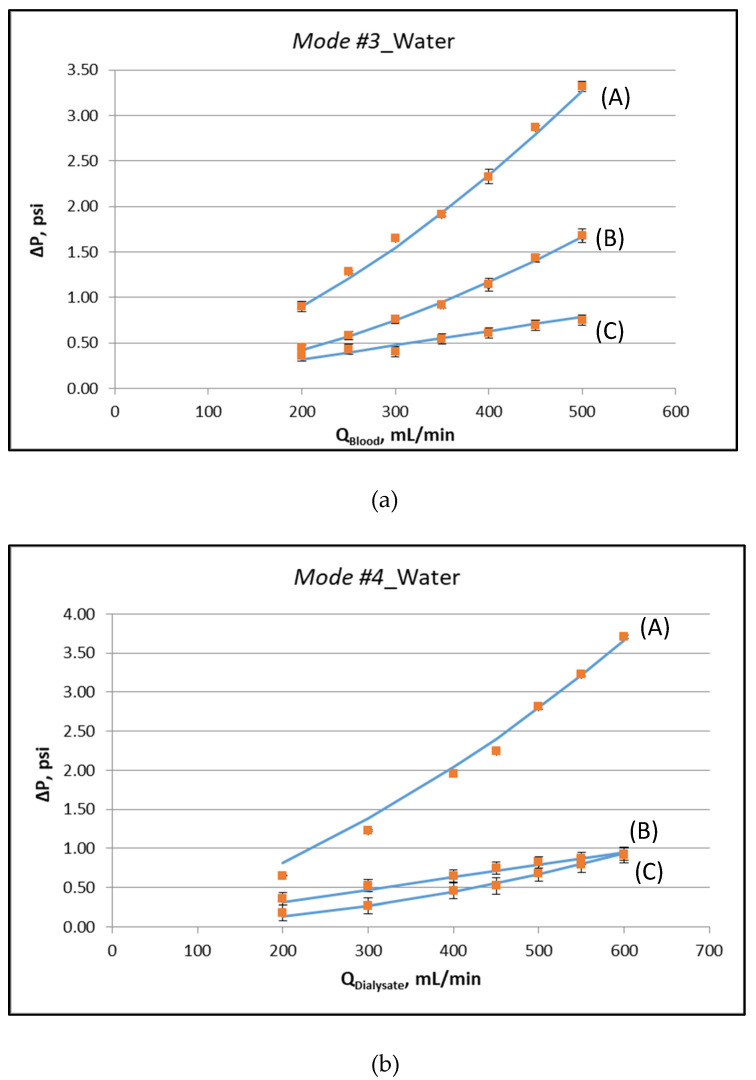
Determination of fluid-mechanical parameters by appropriate data fitting [12], using external pressure differences, for operating *Mode #3* and *Mode #4*. (**a**) Operating Mode #3: (A) ΔP = P_1_ − P_4_, (B) ΔP = P_1_ − P_2_, (C) ΔP = P_2_ − P_3_, (**b**) Operating Mode #4: (A) ΔP = P_3_ − P_2_, (B) ΔP = P_3_ − P_4_, (C) ΔP = P_4_ − P_1_. Detailed experimental data are listed in the Appendix A.

**Figure 4 membranes-12-00062-f004:**
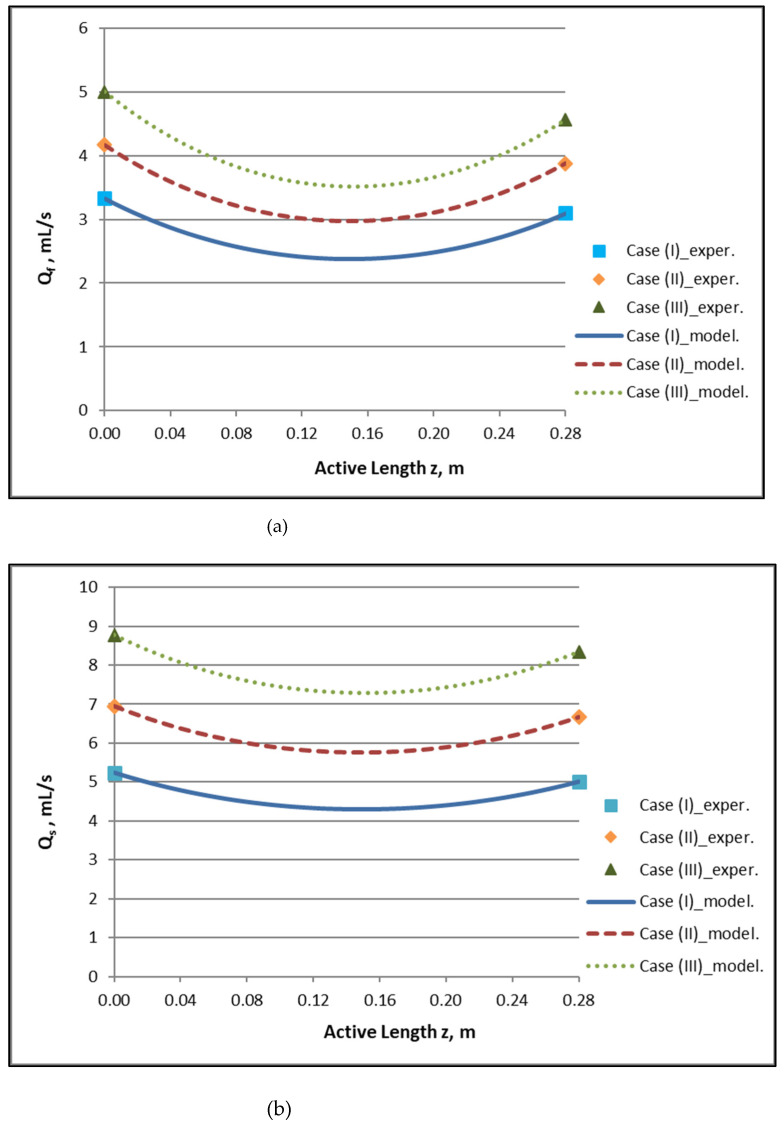
Predicted axial variation of: (**a**) lumen-side flow rate Q_f_/Q_Blood_ and (**b**) shell-side flow rate Q_s_/Q_Dialysate_ for the three cases studied. Case (I): Q_Blood_/Q_Dialysate_: 200/300 mL/min. Case (II): Q_Blood_/Q_Dialysate_: 250/400 mL/min. Case (III): Q_Blood_/Q_Dialysate_: 300/500 mL/min. Data points (■, ◆, ▲) designate imposed/measured inlet flow rates.

**Figure 5 membranes-12-00062-f005:**
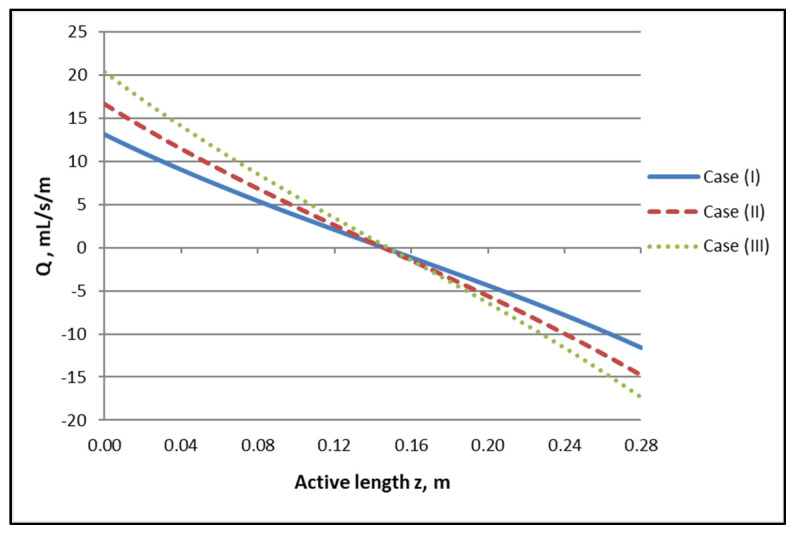
Determined axial variation of local trans-membrane flow rate per unit of length, Q, for the three studied cases. Case (I): Q_Blood_/Q_Dialysate_: 200/300 mL/min. Case (II): Q_Blood_/Q_Dialysate_: 250/400 mL/min. Case (III): Q_Blood_/Q_Dialysate_: 300/500 mL/min.

**Figure 6 membranes-12-00062-f006:**
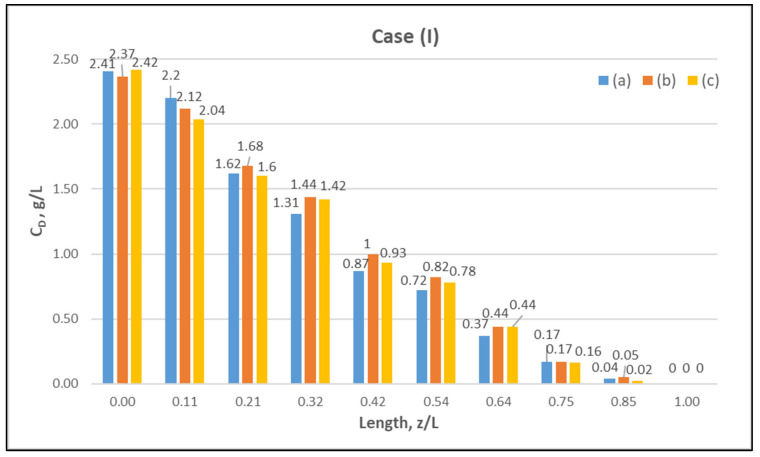
Measured (in triplicate—tests a, b, c) local urea concentration C_D_ on the shell side of the module. Case (I): flow rates Q_Blood_ = 200 mL/min, Q_Dialysate_ = 300 mL/min.

**Figure 7 membranes-12-00062-f007:**
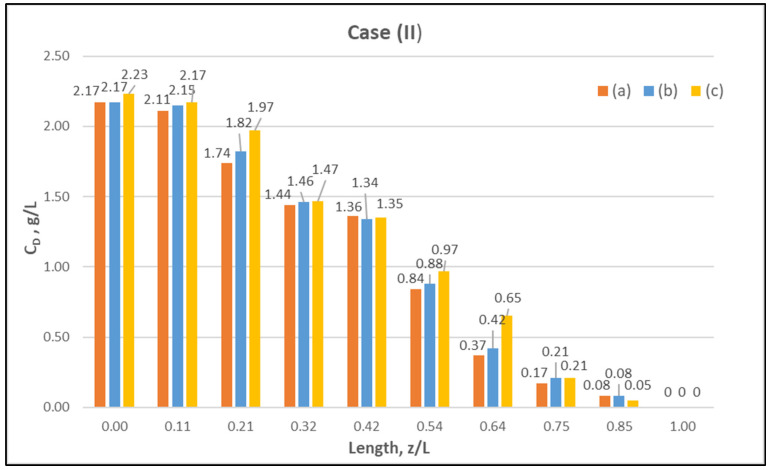
Measured (in triplicate) local urea concentration in the shell side of the module. Case (II): flow rates Q_Blood_ = 250 mL/min, Q_Dialysate_ = 400 mL/min.

**Figure 8 membranes-12-00062-f008:**
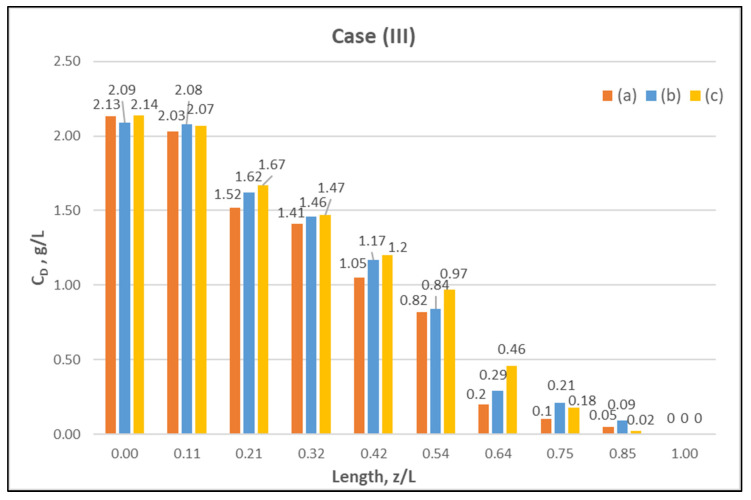
Measured (in triplicate) local urea concentration C_D_ in the shell side of the module. Case (III): flow rates Q_Blood_ = 300 mL/min, Q_Dialysate_ = 500 mL/min.

**Figure 9 membranes-12-00062-f009:**
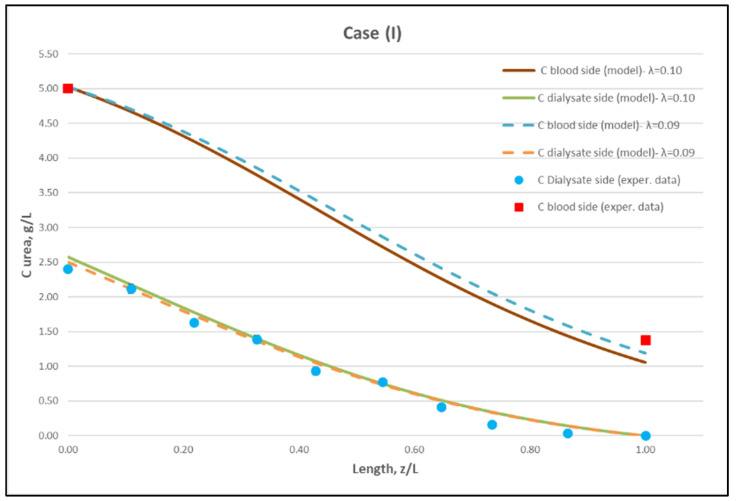
Comparison of measured local urea concentration C_D_ in the dialysate-side, with model predictions, at flow rates Q_Blood_ = 200 mL/min, Q_Dialysate_ = 300 mL/min (Case I). The predicted blood-side urea concentration C_B_ profile is also presented. The points (● and ■) designate measured values of urea concentration.

**Figure 10 membranes-12-00062-f010:**
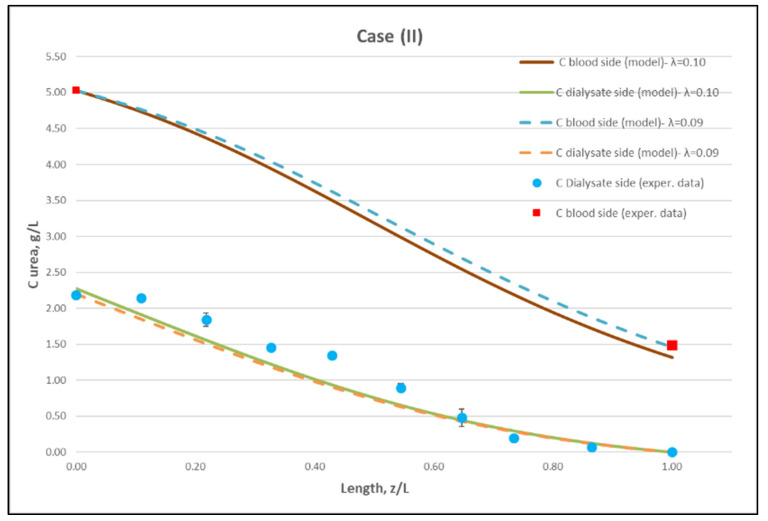
Comparison of measured local urea concentration C_D_ in the dialysate-side, with model predictions, at flow rates Q_Blood_ = 250 mL/min, Q_Dialysate_ = 400 mL/min (Case II). The predicted blood-side urea concentration C_B_ profile is also presented. The points (● and ■) designate measured values of urea concentration.

**Figure 11 membranes-12-00062-f011:**
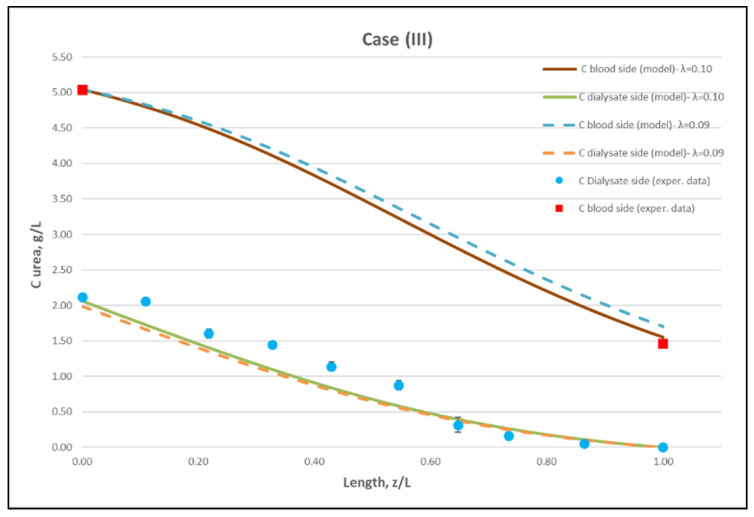
Comparison of measured local urea concentration C_D_ in the dialysate-side, with model predictions, at flow rates Q_Blood_ = 300 mL/min, Q_Dialysate_ = 500 mL/min (Case III). The predicted blood-side urea concentration C_B_ profile is also presented. The points (● and ■) designate measured values of urea concentration.

**Figure 12 membranes-12-00062-f012:**
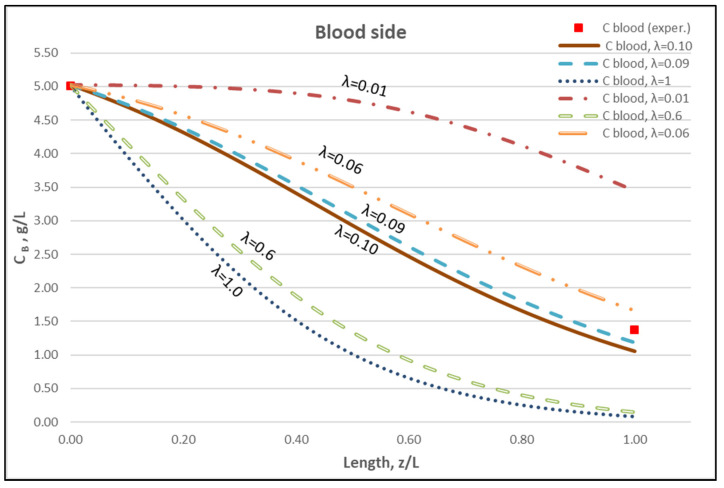
Blood-side urea concentration profile. Influence of the urea effective diffusion coefficient in the membrane (D_e_) on urea mass transfer, under simultaneous trans-membrane liquid convection and solute diffusion. Case (I): Q_Blood_ = 200 mL/min − Q_Dialysate_ = 300 mL/min.

**Figure 13 membranes-12-00062-f013:**
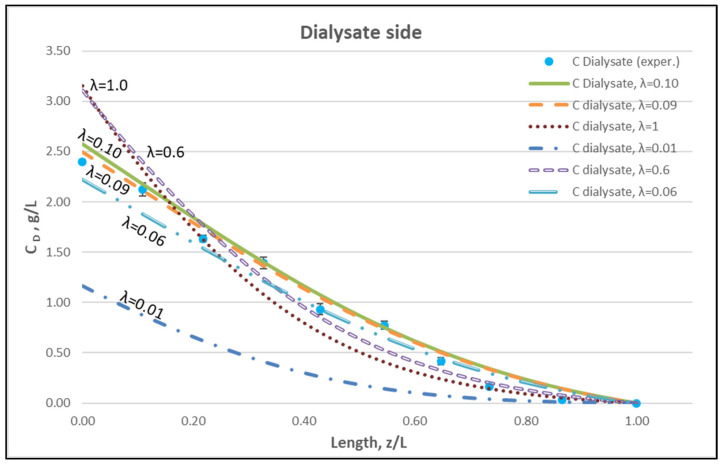
Dialysate-side urea concentration profile. Influence of the urea effective diffusion coefficient in the membrane (D_e_) on urea mass transfer, under simultaneous trans-membrane liquid convection and solute diffusion. Case (I): Q_Blood_ = 200 mL/min − Q_Dialysate_ = 300 mL/min.

**Figure 14 membranes-12-00062-f014:**
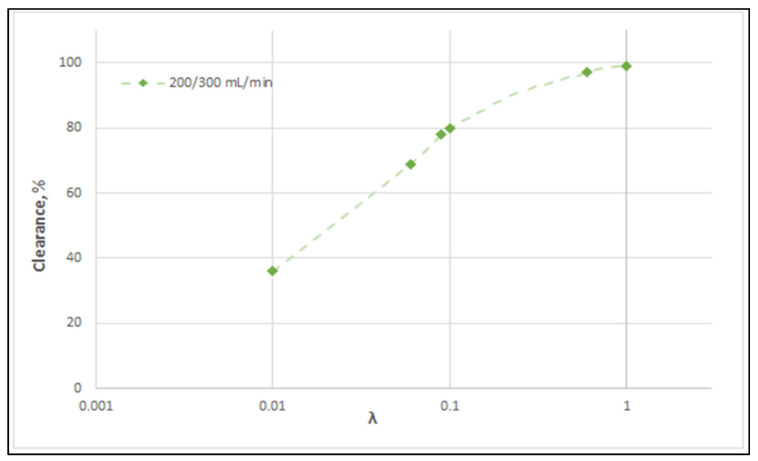
Effect of diffusion hindering parameter λ values on urea clearance for the tested case (I): Q_Blood_ = 200 mL/min − Q_Dialysate_ = 300 mL/min. For values λ → 1, the asymptotic clearance value 100% is reached.

**Table 1 membranes-12-00062-t001:** List of determined parameter values from fitting the model to experimental data.

Parameter	Units	Value
K	mL/(h∙m∙mmHg) m^2^/Pa/s	2.21 × 10^3^ 4.60 × 10^−9^
f_f_	(Pa∙s)/m^4^	5.68 × 10^9^
f_s_	(Pa∙s)/m^4^	0.90 × 10^9^
ζ_1_	Pa/(m^6^/s^2^)	10.25 × 10^13^
ζ_2_	Pa/(m^6^/s^2^)	6.95 × 10^13^
ζ_3_	Pa/(m^6^/s^2^)	5.46 × 10^13^
ζ_4_	Pa/(m^6^/s^2^)	6.88 × 10^13^

**Table 2 membranes-12-00062-t002:** Input parameter values for mass transfer simulations.

Parameter	Symbol, Units	Value
Inner fiber radius	R_o_, m	1.0 × 10^−4^
Outer fiber radius	R_1_, m	1.4 × 10^−4^
Void fraction, shell-side	ε	0.51
Length of active section	L, m	0.28
Number of fibers	N	10,760
Membrane permeance	K, m^2^/Pa∙s	4.6 × 10^−9^
Lumen-side friction coefficient	f_f_, (Pa∙s)/m^4^	5.68 × 10^9^
Shell-side friction coefficient	f_s_, (Pa∙s)/m^4^	8.98 × 10^8^
Diffusion coefficient of urea, at 25 °C	D, m^2^/s	1.34 × 10^−9^
Diffusion hindering factor *	λ	~0.095 *

* Value for best data-fitting.

**Table 3 membranes-12-00062-t003:** Comparison of predicted and experimentally determined urea clearance.

λ	Clearance C_L_ % (Prediction)	Clearance C_L_ % (Experimental)	Clearance K_CL_ mL/min (Predict.)	Clearance K_CL_ mL/min (Exper.)
CASE (I): Q_Blood_ = 200 mL/min, Q_Dialysate_ = 300 mL/min				
0.09	78	74.17 ± 0.18	156	149.3
0.10	80		161	
CASE (II): Q_Blood_ = 250 mL/min, Q_Dialysate_ = 400 mL/min				
0.09	73	72.46 ± 0.66	182.4	181.3
0.10	76		189	
CASE (III): Q_Blood_ = 300 mL/min, Q_Dialysate_ = 500 mL/min				
0.09	69	73.03 ± 0.38	207	220.7
0.10	72		216	

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
