# Peer review of "Mass Transfer Characteristics of Haemofiltration Modules—Experiments and Modeling"

_membranes, 2022, doi:10.3390/membranes12010062_

Round 1

Reviewer 1 Report

The manuscript "Mass transfer characteristics of haemofiltration modules - experiments and modeling" presents model extension and experimental validation for reliable module simulation in Newtonian-liquid flow. The authors described the haemofilter/haemodalyzer fluid-mechanical model and its extension to mass transfer, and as experimental work, they evaluated the mass transfer of urea in the typical counter-current flow mode of hollow fiber. The manuscript is well written and well presented, with good criticism of recent data in the literature. Using fluid mechanical parameters, the concentration profiles of urea, experimentally collected were assessed by a realistic theoretical model. Moreover, it has been demonstrated as the diffusion hindering factor λ, of solute species through the membrane, is the only adjustable parameter in the data assessment and model validation presented herein that represents an intrinsic membrane/solute property that has to be determined experimentally.

The manuscript is suitable for publication in Membranes after minor revision.

1- In the introduction, in the sentence from line 79 to line 83, the authors should highlight/clarify that the described following stages (1- employing the Newtonian human plasma and the effect of membrane fouling and oncotic pressure; 2- extending/adapting the method to haemofiltration of blood) are future perspectives and not data analysed in this manuscript.

2- In figure 3 the legend of operating mode A, B and C is missing. In the figure legends should be added the reference table for P values.

Reviewer 2 Report

This problem is relevant for journal scope. The concept and aim are clearly defined. The content of the paper is well-structured.

I suggest the minor revision of the manuscript.

Remarks and suggestions

  1. Please cite more papers at the last two years in the similar topic of this research.
  2. The main remark: please emphasize the novelty side(s) of your work.
  3. Please add information about mass- and component balance.
  4. Please add the effective area of the applied membrane.
  5. I suggest to add the pressure values in SI unit.
  6. I suggest to revise the figures with few experimental points (Figure 4; Figure 9, 10, 11 with red points).

Reviewer 3 Report

This study provides valuable insights into the haemofiltration process through a sensitivity analysis involving key process parameters. Significant results are also obtained regarding the relative contributions of convective and diffusive mass-transfer. Overall, the experimental design is reasonable and most of the conclusions can be proved. However, the manuscript as it is written now, needs to be further improved. Particularly, the authors should address the following points:
1.It is recommended that paragraph format should be aligned to both ends to avoid hyphens use.

2.Is ultrafiltration model used for haemofiltration different from other ultrafiltration mass transfer models? As we know that ultrafiltration is a membrane process very similar to reverse osmosis. A simple comparison of the model with other mass transfer models is recommended in the introduction, and relevant mass transfer models have been reported by Chemosphere 269 (2021) 128686; Processes 2019, 7, 271, etc. Please review and cite these existing reports in introduction.

3.Page 3, line 114-119: It is recommended that the serial numbers after the formula be aligned, as are the other formulas in this article.

4.Page 6: Figure 1 only shows the data of Z model when γ=1, 2, 5, 0.5. Although the results of Z model when γ=3 are similar to KK2 model, there are still some differences. It is suggested to draw the line of γ=3 in the figure to make the comparative analysis more intuitive.

5.Page 7, line 281: The actual operating temperature of hemodialysis is generally set close to the human body temperature of 36.5℃, and the experimental temperature is 25℃, whether this temperature has an impact on the experimental results? Urea solution is used as the blood side feed solution, but the blood substance composition is complex and the viscosity is larger than urea solution.It may not be able to fully reflect the actual situation.

6.Page 6: In figure 3, it is suggested that three fitting curves should be marked respectively.

7.Page 15: The formats of ff and fs in Table 2 are incorrect. “f” and “s” are subscript formats.
